# “Climate Change and Health?”: Knowledge and Perceptions among Key Stakeholders in Puducherry, India

**DOI:** 10.3390/ijerph20064703

**Published:** 2023-03-07

**Authors:** Shreya S. Shrikhande, Sonja Merten, Olga Cambaco, Tristan Lee, Ravivarman Lakshmanasamy, Martin Röösli, Mohammad Aqiel Dalvie, Jürg Utzinger, Guéladio Cissé

**Affiliations:** 1Swiss Tropical and Public Health Institute, Kreuzstrasse 2, 4123 Allschwil, Switzerland; 2Faculty of Science, University of Basel, 4001 Basel, Switzerland; 3State Surveillance Officer, Department of Health and Family Welfare Services, Government of Puducherry, Puducherry 605001, India; 4Centre for Environmental and Occupational Health Research, School of Public Health and Family Medicine, University of Cape Town, Cape Town 7925, South Africa

**Keywords:** cardiovascular disease, climate change, health, India, key informant interviews, low- and middle-income countries, qualitative study, stakeholder perspectives

## Abstract

Climate change has far-reaching impacts on human health, with low- and middle-income countries, including India, being particularly vulnerable. While there have been several advances in the policy space with the development of adaptation plans, little remains known about how stakeholders who are central to the strengthening and implementation of these plans perceive this topic. We conducted a qualitative study employing key interviews with 16 medical doctors, researchers, environmentalists and government officials working on the climate change agenda from Puducherry, India. The findings were analysed using the framework method, with data-driven thematic analysis. We elucidated that despite elaborating the direct and indirect impacts of climate change on health, there remains a perceived gap in education and knowledge about the topic among participants. Knowledge of the public health burden and vulnerabilities influenced the perceived health risks from climate change, with some level of scepticism on the impacts on non-communicable diseases, such as cardiovascular diseases. There was also a felt need for multi-level awareness and intervention programmes targeting all societal levels along with stakeholder recommendations to fill these gaps. The findings of this study should be taken into consideration for strengthening the region’s climate change and health adaptation policy. In light of limited research on this topic, our study provides an improved understanding of how key stakeholders perceive the impacts of climate change on health in India.

## 1. Introduction

To paraphrase an environmentalist we interviewed for this study, “*Climate change is like a ticking time-bomb that is going to blow up in the near future in a way we cannot even imagine.*” Across the globe, evidence of the changing climate and its far-reaching impacts on our world, including peoples’ health and well-being, have been reported, with the Lancet Commission declaring it to be the biggest threat to global health [1,2,3].

As an added factor, climate change compounds on top of existing health vulnerabilities for several climate-sensitive outcomes, including vector-borne diseases, water-borne diseases and non-communicable diseases (NCDs), such as cardiovascular diseases (CVDs) [2]. Given the trends projected for climate change, these risks are expected to increase in the future. By 2050, an excess of 250,000 deaths per year are projected to be attributable to climate change, through several exposure pathways, ranging from heat stress and changes in vector-breeding patterns to malnutrition caused by changes in agricultural productivity [2,4].

The impacts of climate change are region specific, with unequal distribution both within and between countries, regions and even communities [5]. There is evidence of effective and timely adaptation practices reducing or potentially avoiding several climate associated health risks [2]. A growing number of countries have committed to developing national climate adaptation or action plans, especially after the Paris Agreement in 2015 [6,7]. However, the regional differences in vulnerabilities and subsequent impacts remains a key challenge for developing these plans. National and regional indicators like socio-economic factors, access to healthcare, infrastructural developments and even awareness determine individual vulnerability to climate change, especially in resource limited low- and middle-income counties (LMICs) such as India [8,9]. While there have been several advances in this sphere in India, there remains great scope to comprehensively enhance and strengthen health adaptation strategies and health determinants, thereby improving public health [10,11,12]. These efforts are often undermined by ineffective climate adaptation policies and strategies [13].

In light of these vulnerabilities and gaps, there is a need to explore the perceptions of key stakeholders, such as policy makers, researchers, environmentalists and medical staff, on the health impacts of climate change in India. Policy makers, including ministerial officers and scientific researchers play a key role in generating evidence, designing, implementing, monitoring and evaluating climate change-health-related policies and plans. Although there is limited research on this topic, the knowledge, perceptions and awareness among these groups of people have been shown to influence political prioritisation of climate change adaptation [14,15]. Another group of key stakeholders includes public health professionals. A recent literature review showed that public health experts, despite being aware of the consequences of climate change, lack sufficient knowledge on its health impacts [16]. As first responders, healthcare providers play a key role in identifying climate impacts on health, recognising opportunities for health-in-climate actions, advocating for policy support to address climate impacts on health and building climate resilient healthcare systems [17]. However, the extent of their work is largely dependent on public health policies or adaptation frameworks in place to support them [18].

India is characterised by a high disease burden and climate change is thought to further exacerbate vulnerabilities [19,20,21]. There is a need for multi-factorial studies on this topic, including qualitative studies, as little is known about the perceptions of stakeholders with regard to the nature, magnitude and urgency of climate change impacts on peoples’ health in India. Hence, it is important to examine this nexus in order to facilitate a region-specific understanding of how key stakeholders involved in the field of climate change and/or health perceive these impacts. This exploratory study has the potential to contribute towards addressing policy gaps and tailoring specific interventions to reduce the health burden attributed to climate change in India. Against this background, we aimed to understand how the health impacts of climate change are perceived by two key stakeholder groups in Puducherry, namely those working in the health sector and those involved in environmental planning and policy development. We also chose to include questions that focused specifically on CVDs, the diseases with the largest burden in India, along with other health impacts [22]. The findings from this research feed into a larger project examining the impacts of climate change on CVDs in Puducherry. The objective was to identify the knowledge and awareness, the perceptions of climate change health links, key gaps in knowledge and services, and scope for improvement of future climate change health policies and plans.

## 2. Materials and Methods

### 2.1. Study Setting

Puducherry is a unique Union Territory (UT) in India, comprised of four erstwhile French colonies (i.e., Puducherry District; Karaikal; Mahe; and Yanam region) in the south-eastern part of India (Figure 1A). The total area of the UT is 492 km^2^.

This study focuses on Puducherry district, which itself covers an area of 294 km^2^ spread out over four non-continuous sub-districts or ‘Taluks’ (Figure 1B). As per the Government of India census of 2011, the population of Puducherry is 950,289 with an almost equal distribution of males and females [23,24]. Puducherry falls within the tropical savannah with a dry winter climate type as per the Köppen-Geiger classification, with a mean annual temperature around 30 °C and a temperature range from 23 to 41 °C. A study describing the causes of death in Puducherry for the first time found that 31% of deaths were attributed to CVDs followed by external causes, digestive diseases, respiratory illnesses and infectious diseases, respectively, providing a glimpse into the health profile of the region [25].

### 2.2. Study Design and Participants

We conducted key informant interviews (KIIs) with medical professionals (both in practice and research) and officials from the Department of Science, Technology and Environment (DSTE) involved in the Puducherry State climate change action plan.

Participants were initially selected using purposive sampling based on professional relevance to the topic and prior connections. To that end, the sample included in this study consisted of interviewees with either a medical or an environmental professional profile. Further interviewees were invited through snowball sampling [26]. We specifically chose this method to ensure that we were able to elucidate the perspectives of professionals working closest to this field and those who would be able to provide us with the most granular information. We continued to recruit key informants and carried out interviews until saturation was reached in the information provided. In the case of the DSTE representatives, we interviewed all possible relevant participants. Informant recruitment was also carried out with practical considerations, bearing in mind COVID-19 restrictions and schedules of the medical professionals.

Out of the informants, all but five were medical professionals either actively practicing medicine or working in academia. For the medical professionals, we focused mainly on trained cardiologists, emergency medicine and general medicine doctors with varying levels of experience. The majority of doctors interviewed were male, with only one female informant. The entire sample only contains three female participants. The participants’ profiles are shown in Table 1.

### 2.3. Data Collection

A total of 16 interviews were conducted in Puducherry between January and March 2022 with stakeholders working in this region. Participants were invited for an in-person interview at their place of work at a convenient time. For two participants, who were unavailable for an in-person interview, virtual interviews were conducted over Zoom. Interview questions were based on an a priori, developed, semi-structured interview guide based on the objectives and analytical framework. It included general questions on climate change and health asked to all participants, as well as profession-specific questions. The conceptual framework for climate change risk perceptions developed by van Eck et al. [27] and the framework for health inequalities proposed by Rudolph et al. [28] was used as a base for our analytical framework (Figure 2). This paper focuses on the themes of ‘systems knowledge’ and ‘socio-cultural dynamics and public engagement’. Findings pertaining to the theme of ‘institutional determinants’ are discussed elsewhere.

The guide was finalized following three pilot interviews, but was purposefully kept flexible depending on the responses of the participant. The interview guide is presented in the Appendix A. All the interviews were carried out in English by the first author (S.S.S.), while another author (R.L.) facilitated the interviews and was present for three interviews as a passive observer. There was no prior relationship between the interviewer and interviewee. At the start of each interview, the interviewer gave a brief personal introduction and explained the informed consent form (ICF), which included a detailed description of the study objectives, the role of participants and planned output. A copy of this form was provided to all participants with contact details of the study team. Interviews lasted for approximately 15 to 50 min, with the average interview lasting around 30 min, and were audio recorded using a simple voice recorder. Field notes were also taken during the interview to aid the optimisation of the interview guide, for key issues and themes that emerged.

Broad and open-ended questions were asked to allow participants to freely express thoughts and perspectives and discuss about any issues on their mind not explicitly asked through questions. The structure and order of questions varied based on discussion points but roughly covered the same key topics. Every effort was made to minimise the impact of potential biases throughout the study by keeping note of and reflecting on them throughout the interview, analysis and write-up phases. Participants had an option of requesting a recording, which was requested by one participant. The final question included participants’ comments about the study, which were taken into consideration while adapting the questionnaire.

### 2.4. Data Analysis

The recordings were assigned a number before verbatim transcription by the first author (S.S.S.) to ensure anonymity. Transcription and analysis were completed using MaxQDA software version 2018.1 (VERBI Software, Berlin, Germany)).

We used a combination of deductive and inductive thematic analysis following the framework method developed by Gale et al. [29]. We first developed broad codes and themes based on our framework and interview guide. Further codes were developed inductively through open coding during the analysis to identify emerging themes and additional categories, which were then clearly defined in our framework.

After familiarisation with the transcripts, the interviews with the richest data quality were used to develop the initial codebook and analytical framework. These included one interview from a medical doctor, a medical researcher and an environmentalist. The codebook was validated by two authors (S.S.S. and T.L.) in an effort to address discrepancies and agree on the final coding system, which was then applied to the remaining transcripts. The codebook was updated to include any themes not covered in the initial three interviews during the inductive coding to develop a final framework matrix. All transcripts were re-read at the end of the first round of coding to cover any missing or additional codes. Summary sheets were developed by code to better understand the themes that emerged within and across interviews. Relevant quotes were then directly charted into our framework matrix covering two themes and nine categories. The structural codebook and framework matrix is presented in the Appendix A. This was further elaborated using a cross-matrix, also developed using MaxQDA.

### 2.5. Ethical Considerations

Written informed consent was obtained through the ICF prior to the interview, which included the participant’s rights to withdraw from the study at any stage without further obligations. The participants were provided with contact details of all the researchers involved in the project. All quotes form stakeholders presented here are assigned a serial number to further ensure anonymity. Additional relevant quotes are presented in Section 3 of the Appendix A.

This study was approved by the Institute Ethics Committee (Human Studies) of the Indira Gandhi Medical College and Research Institute (a Government of Puducherry Institution); reference no. 318/IEC-31/IGM&RI/PP/2021 and by the Ethics Committee Northwest and Central Switzerland (EKNZ); statement ID- AO_2020_00034. The methodology used in this project abided by the principles laid out in the Declaration of Helsinki and the COREQ checklist.

## 3. Results

We presented our findings based on two main thematic areas, namely the understanding of climate change and health (systems knowledge) and socio-cultural factors that influence public awareness and engagement on this topic (sociocultural dynamics and public engagement).

### 3.1. Systems Knowledge

#### 3.1.1. Climate Change Is an Acute and Growing Problem for India

All participants were aware of, but had varying understanding of climate change. Extreme heat, increase in urban flooding, changes in monsoon patterns and air pollution were some of the terms most commonly associated with climate change, along with plastics, global warming, climate refugees and droughts. One participant also mentioned how “*climate refugee is going to be the word of the century”.* Many participants also directly associated climate change with “*reducing the quality of life”*, mainly through extreme weather events, and expressed concern about the growing burden of climate change-related issues. Participants understood the within- and between-country differences in climate impacts and thought of it as an acutely and disproportionately growing burden, especially for the future.

“ *…climate change is extremes of weather in many cities across the world, including Delhi and other parts of India which is making human beings suffer*”.(#15, practicing physician)

Participants also described dystopic views about the future due to a rapidly changing climate.

“*When I think of climate change, it gives me a very gloomy picture about the future*”.(#7, Environmentalist)

“*In 20 years, or by the end of the century, it is going to create a huge problem that by then nobody could tackle or could come over*”.(#9, Environmentalist)

Participants cited diverse sources of climate knowledge and awareness, including media and international organizations as well as personal observations. Some of the environmentalists also backed up their thoughts with results from climate change-related studies they conducted. The most strongly stressed observation by the environmentalists was the relatively recent acceleration of climate change. This sentiment was echoed by almost all participants, who described a remarkable change in weather patterns over the past two decades. Phrases such as “*as the years passed, climate has changed so much*”, and “*what I saw as a child in the 1980s, like there we used to get, uh, showers fed throughout the monsoon season. But now it’s all sudden downpour on a single day*” were used to describe the change over recent time. Participants described observing dramatic changes in weather patterns, especially over the past 10 years.

“*I’m in Pondicherry since 1989, the initial 10 to 15 years things were very stable. For the past 15 years, things have been much worse. The things are becoming more severe. So, temperatures are becoming higher in Pondicherry and we are having a lot of cyclones and everything in the past 10 years compared to the first 20 years I was in Pondicherry*”.(#15, Practicing physician)

The unpredictability of seasonal patterns and heat were also mentioned multiple times to support the severity of the problem. Many mentioned the prevalent heat throughout the year, transcending the erstwhile seasonal boundaries, along with harsher and irregular monsoons.

“*Pondicherry, it’s not hot season- cold season, it’s hot, hotter, hottest. So that’s the change, so we cannot tell it’s cold season…Even during the rains or other things, we feel that the background heat is there and it looks like summer throughout*”.(#4, Medical doctor/researcher)

#### 3.1.2. Climate Change Ultimately Affects Health through Domino Effects

The awareness of the impacts of climate change were largely informed by day-to-day experiences and observations for most participants. Several participants described the interconnected chain of impacts they attributed to climate change. Quality of life and agricultural impacts arising from changes in monsoon patterns were frequently mentioned. Regardless of the sector directly impacted, most participants perceived health to ultimately be impacted through a chain of cyclical or downstream effects of climate change.

“*Because, you know, ultimately, all sectors will lead to health of the individual. Let it be financial, let it be agriculture, let it be electricity, let it be road, whatever, it’s going to affect the health of the people*”.(#2, Medical doctor/researcher)

An example of this domino effect was unseasonal weather patterns causing agricultural disturbances and leading to other problems, such as unemployment, economic problems, disruption of the biological natural cycles and eventually health, including malnutrition. Overall, health was mentioned by almost all participants as being directly or indirectly affected by climate change. One Health as a concept was also mentioned by a medical researcher and described by a doctor and environmentalist with the belief that climate adaptation strategies needed to be framed from a One Health lens.

*“I strongly believe in the concept of One Health. So there’s like this impact on everything. So if one gets tilted, it’s like a domino effect. It affects every part of it…So starting from agriculture or animal health, human health, everything gets largely affected”*.(#4, Medical doctor/researcher)

#### 3.1.3. Knowledge of the Public Health Burden and Vulnerabilities-Influenced-Perceived Health Risks from Climate Change

Vector-borne diseases, such as malaria, dengue and chikungunya, were considered to have the greatest burden from a public health perspective in Puducherry. These were also the most commonly mentioned group of diseases perceived to be affected by climate change, largely based on local experiences, observations and public health experiences in dealing with them.

“*Primarily, we deal with a lot of VBDs [vector-borne diseases]. What we say is, as the global warming is going to affect more and more areas, the mosquitoes and other vectors are going to breed horizontally and vertically, they’re going to expand horizontally as well as vertically*”.(#2, Medical doctor/researcher)

Altered breeding patterns of mosquitoes in recent years and new reported incidences of malaria in hilly regions not prone to outbreaks were examples given to support the view of climate change increasing the breeding window for mosquitoes. Changes in monsoon patterns and increased incidences of flooding were also considered to foster new suitable breeding grounds for mosquitoes.

While there was some awareness on other climate-sensitive diseases, such as NCDs, most considered them to be primarily affected by other risk factors, predominantly lifestyle. A few participants thought of respiratory illnesses such as chronic obstructive pulmonary disease and asthma as climate sensitive. Mental health, seasonal and heat-related illnesses such as dehydration, skin diseases, allergies, CVDs and heat stroke were mentioned by a few participants, mainly doctors, as climate-sensitive.

Factors such as gender and occupation were considered important drivers of climate vulnerability and its subsequent health impacts. An interesting observation made by one of the medical researchers, pointing to an often underrepresented gender-based occupational hazard and vulnerability is described:

“*Working women actually will be a problem because many of these construction workers and people who are working in this sector, even shops and other things you know, they have a difficulty-they don’t have a privacy for using the restroom. So they don’t drink water much thinking that if they drink water, they have to search for the restroom, which is not available. So they don’t hydrate themselves*”.(#4, Medical doctor/researcher)

Across the board, the aged and socio-economically disadvantaged populations were thought to be most vulnerable to adverse health outcomes associated with climate change. There was no consensus on different vulnerability levels between urban and rural populations, with several participants considering socio-economic factors as more important determinants of vulnerability. The majority of respondents mentioned those engaged in outdoor work and the unorganized sector such as agriculture, fishing and construction as those being more vulnerable to climate impacts. Socio-economically disadvantaged people were thought more likely to disregard health concerns, especially in the early stages, often delaying treatment. Health complaints were also thought to likely be brushed off as occupational hazards, with livelihood coming at the cost of well-being.

“*A lot of research has shown that you know, health seeking behaviour depends on the felt need. So we have seen that you know, the lower economic strata may assume many things to be normal because they’re more of manual labourers. So they may not be very much worried about a fever or a headache*”.(#2, Medical doctor/researcher)

#### 3.1.4. Indirect Health Impacts of Climate Change Experienced

Some respondents described feeling powerless in the face of climate-related damages, which have long-lasting impacts on life, including on mental health. Extreme weather patterns and changes were perceived to be an important driver for unemployment in the agriculture sector. One environmentalist discussed the difficulties workers have in being motivated to re-join the workforce or finding gainful employment due to lack of demand, thus affecting their mental health. This was also linked to laid-off workers becoming increasingly prone to NCDs through a sedentary lifestyle.

“*For some (because of) the climate change, they don’t have agricultural work… and they become sedentary*”.(#6, Practicing physician)

In conjunction with easily accessible and relatively cheap fast food, several participants mentioned the changes in food patterns, ranging from downstream effects of insufficient harvests, lifestyle modifications and increased prices of food. This was also largely perceived to contribute to the development of health issues.

Most of the environmentalists and a few medical professionals also mentioned the impacts on water quality, from an increase in salinity to groundwater depletion. Salinity seemed to be considered as a problem for Puducherry, being a coastal region, with one of the medical researchers describing ongoing studies on the association between salinity and hypertension. Other participants, particularly the environmentalists, also described health issues they had faced due to contaminated groundwater and flooding. The challenges in ensuring a continued adequate supply of clean water were also highlighted, with one environmentalist expressing concern about competition for water in the future.

Air pollution, while thought of as a factor affecting health overall, was not seen as a cause of concern for Puducherry. Multiple participants stressed on the relatively clean air in Puducherry, with one environmentalist describing the results of DSTE study on air pollution. Some participants also highlighted air pollution monitoring displays in the city as their source of awareness on air quality in Puducherry.

“*The only thing that we are not facing an issue, unlike Delhi or Mumbai or other places is that this sea breeze is taking away things. The dispersion is quite good here; I have also carried out the air pollution studies for the last 15 years data…Not even close to an alarming situation, because of our geographical gift or whatever*”.(#9, Environmentalist)

#### 3.1.5. Scepticism about Climate Change Affecting CVDs

As part of a larger study, we wanted to focus on the participants’ views on the impacts of climate change on CVDs. We found divergent opinions on whether and how climate change, especially temperature, affects CVDs. Although some mentioned CVDs to be temperature sensitive, especially to extreme heat, others also voiced the belief that the scientific community does not know much about the association between climate change and CVDs. Most environmentalists believed temperature, mainly heat, to be a risk factor for CVDs, but believed vector-borne diseases to ultimately be more affected.

While CVDs are mentioned as one of the heat-sensitive diseases in the National Action Plan for Climate Change (2008) [30] and the National Action Plan on Climate Change and Human Health (2018) [31], almost all participants felt they did not have enough data to confidently inform their opinions. The prevalent belief is that CVDs are largely a multifactorial, lifestyle-linked group of diseases; the long development time, and subsequent difficulty in isolating the environmental risk factors were thought to contribute to this uncertainty. Some participants proposed the indirect link between climate change and CVDs through changing food habits and stress caused by climate uncertainty.

Other participants expressed scepticism about the association between climate change and CVDs at large or within Puducherry, partially due to insufficient knowledge or awareness about current research on the topic.

“*For example, hypertension, dyslipidaemia cause heart disease. There is direct evidence, multiple meta-analysis and everything was there. But there is no…say like no study categorically saying that climate change will cause cardiac disease…as of now, I’m not convinced about the cardiovascular disease and the climate*”.(#13, Practicing physician)

“*At the present point, we cannot tell whether increase in temperature will cause so and so heart disease. Without that link, we cannot prepare now. See high temperature—if at all I’m correct, like it can cause a heat stroke. But I have not read anywhere, like sudden rise in temperature causing a cardiac disease*”.(#14, Practicing physician)

This was supplemented by an expressed difficulty by doctors in identifying temperature-attributed CVD cases in hospitals, partially due to the relatively stable temperatures in Puducherry without extreme heat or cold.

“*See I don’t think such things will happen in Pondicherry. We don’t have extreme climates out here. See all these things. Maybe north. In the south nothing*”.(#5, Practicing physician)

Interestingly, a few of these same participants classified CVDs as being heat sensitive. This could point to a gap in associating temperature with climate change, as most medical professionals were able to describe thermoregulatory changes in response to heat or cold. However, this finding was limited to the cardiologists, with other participants not displaying much knowledge about the burden of CVDs or their association with environmental factors.

Another perspective was that the focus was on more ‘explicit’ diseases from a public health perspective, such as vector-borne and communicable diseases. While NCDs such as CVDs have an extremely high disease burden, they are seen as the “silent” disease commonly associated with personal lifestyle choices, as opposed to external risk factors. One medical researcher also described the overwhelming onus on individual choices when it came to CVDs as opposed to external influences, leading people and policies to not prioritise them. As one cardiologist put it:

“*Awareness regarding cardiovascular diseases is the number one cause of death itself is not there…..nobody in India knows that in 2020, [approximately] 2 million people died of COVID across the world, but 10 times that died because of cardiovascular disease. That is 20 million. Everybody thinks that the most common cause of death in 2020 was COVID. … And before that everybody thinks of cancer, or accidents and terrorism, all these things kill more people, but actually cardiovascular disease kills maximum number of people*”.(#15, Practicing physician and cardiologist)

### 3.2. Socio-Cultural Dynamics and Public Engagement

#### 3.2.1. Perceived Credibility and the Societal Role of Information Communicator Are Important for Uptake and Public Awareness

The relative lack of awareness of climate change and health among the population combined with the perceived need to mobilize the medical community to engage in environmental health literacy was a point of discussion. The lack of awareness among the scientific community on climate change and health was brought up several times. The scientific community, especially the medical community, were seen as leaders and there was a felt need for them to lead the charge on climate change and health-related outreach. The need to train the medical fraternity to recognise climate sensitive diseases was also mentioned, especially as doctors in Puducherry were ranked high on diagnostic skills.

“*It is more of, you know, making the medical fraternity as advocates for this issue. So I think diseases we are very good, you know, we will treat the patients we will do this thing. But the awareness level, you know, even for a scientific community as doctors there is very less regarding climate change*”.(#2, Medical doctor/researcher)

When it came to information sources, the credibility and perceived role in society were considered paramount to how the information would be received by the population and their reaction to it. The more credible a source was deemed to be, by virtue of professional title, education or celebrity, the more likely the audience was thought to able to absorb or trust the information. Doctors, for example, were thought to a trusted source of information through their education and standing in society, leading people to accept the message easier.

“*The same thing the two people can say, but when the doctor says it, they (people) will easily take it. But if I’m saying that (the reaction will be) “Okay, someone is talking” and (people will) leave it. So those who are working in that (area) will connect it easier. If I’m telling you about health, (you) won’t understand and if the working people like doctors or those who are in the research—if they tell, they can easily connect and you will easily understand*”.(#10, Environmentalist)

In line with this, famous personalities such as Narendra Modi and Greta Thunberg were mentioned in relation to climate change, although here political inclinations of the participants influenced how credible they perceived the source to be.

Finally, several participants mentioned newspapers and other media channels, including the daily news, as their primary source of information. Some even mentioned how they turn to these sources for guidelines on issues such as climate change and consider mass media to be a potential medium to spread awareness on climate change and health impacts or preventative measures.

#### 3.2.2. Need for Alternate Solutions and Incentivized, Targeted Programs on All Societal Levels

One of the participants, when speaking about awareness and measures to counter climate change, emphasized the need to provide alternative solutions to everyday behaviours to improve health. This was accompanied by the perceived need for a legal framework or mandate to support public health initiatives, policies and encourage behaviour change. Climate actions in the form of popular campaigns, such as tree plantations, were seen by one participant as a temporary fad with little practical longevity. Issues in maintaining these programmes in the long term were also mentioned.

“*We are advising the people to do this, do this. But the problem is we are not providing an alternative solution for that. …When you’re advising for avoiding use of plastics (for example) what is the alternative we are providing. Without providing alternative, we can’t ask the people to change*”.(#13, Practicing physician)

This was somewhat echoed by another participant when talking about vulnerable populations prioritising economic needs at the cost of their health and thus lacking the incentive and interest to back climate action. This highlights the need to consider the health needs of vulnerable population groups and creating provisions for them when developing climate adaptation guidelines or plans. There were concerns that despite being warned, the most vulnerable population would disregard risks, which needs to be taken into consideration.

“*Even when you give them awareness, they will not be ready to follow all those because they have to take care of their daily living. So, when you say that or ‘do not work more number of hours outside’ or even during any severe heat waves IMD [Indian Meteorological Department] gives warning. As a State, we can give warnings. Severe heatwave is not more than two days a month, but even then, we cannot make people or we cannot compel them to, you know, take care of themselves. We can just give warning*”.(#9, Environmentalist)

The need to tailor adaptation plans, awareness campaigns and programmes to be relevant and accessible to different types and groups of the population was also stressed by several participants. The need for programmes to start from “grassroot level*”* was especially perceived to be imperative by most of the participants. There was a perceived disconnect in the needs of the most vulnerable population and the planned programmes at a higher level. One participant also highlighted having sub-committees focused on the needs of different population groups. The environmentalists also mentioned holding webinars or information sessions tailored to children or farmers, for example, which could also be used as a means to communicate climate change-health awareness in the future.

#### 3.2.3. Integrating Climate Change Impacts in Schools, Universities and Continuing Education Curricula

Nearly all participants were of the opinion that the curriculum needed to be adapted for all educational levels. Participants especially stressed the need to include climate change as a separate module in the medical curriculum. The underlying feeling was that medical students need to be taught about climate impacts on health in order for them to be able to effectively treat and spread awareness among their patients. Topics such as climate impacts on health and health system management in light of resilience were thought to be of importance for inclusion in the medical curriculum. Some participants, who were parents of young children, also expressed a desire to see more content specifically focused on climate change in schools. One even spoke about teaching their child about the importance of climate change from home to make up for the lacunae felt in the primary school curriculum.

Another option discussed was adding a climate component in continuing education courses, especially for those planning to work in climate change or health-related fields. Open access lectures or webinars on climate impacts on health were recommended as a potential solution in order to increase awareness on the topic among the general public. Climate change and health as a topic in scientific conferences were also considered a good way to disseminate current research within the scientific and medical community.

“*Once in six months, some courses of some continuous medical education, like that. If it is conducted for conferences, medical conferences it will be beneficial at present state, once the real burden is more than we can implement in the medical curriculum also*”.(#16, Practicing physician)

Not all participants were open to taking courses, especially doctors, thinking it to be more beneficial for the younger generation; however, all were willing to engage in information sessions or seminars of a short duration.

#### 3.2.4. Seasonal Workplace Guidelines

Related to the aforementioned finding on considering the needs of different population groups in adaptation guidelines, a few participants also mentioned the need to include temperature or season-related workplace and school guidelines. Schools and outdoor workers were especially thought to be the targets for these interventions. Heat in particular was seen as the exposure which people needed protection from. One participant even drew comparisons to similar programmes already practiced elsewhere in India as a benchmark.

“*Policies in the sense that change in the workplace guidelines. Especially people who are working in the open area like road workers and agricultural labourers and other things. So they should bring in some policy change in the sense that when the temperature goes above one particular limit and below one particular level, they should avoid working during daytime and night-time. For example, extreme temperatures, high temperatures, they should not be working between let’s say 11 am and 4 pm. And even schools and colleges and all they close during that period. Same thing with low temperature. … So same thing should be practiced across all offices so that it becomes more bearable for the people*”.(#15, Practicing physician and cardiologist)

A participant also brought up the need for improved early warning and dissemination systems for temperature levels and warnings. On the other hand, contextual challenges in implementing early health warning systems were mentioned by one participant, which require further investigation.

## 4. Discussion

To date, little is known about the perceptions, knowledge and attitudes to health impacts of climate change in Puducherry and elsewhere in India. By focusing on two interlinked groups of stakeholders, namely medical professionals and officials from the DSTE, we attempted to understand the fundamentals of how this topic is perceived from a medical practice and policy lens. Our findings focus on the system’s knowledge about climate change and its health impacts, socio-cultural dynamics, public engagement and recommendations.

Consistent with findings from large-scale, global studies and reviews, all medical professionals were aware of climate change [16]. As was also seen in a study by Sheffield et al. from Harlem, New York, the feeling that ‘things are changing’ in Puducherry was prevalent amongst the participants, especially over the past two decades [32]. This can be supported by evidence of a steady rise in temperatures across southern India in the past few decades with more records of extreme temperatures along the eastern coast in addition to changes in rainfall patterns and increases in cyclonic activity [33].

In line with several other studies on this topic, perceptions and beliefs about health impacts of climate change were shaped by personal experiences and direct observations [32,34,35,36]. This could potentially explain why participants deemed common and easily observed climate risks and health impacts, such as agriculture and vector-borne diseases, greater than others, such as CVD impacts. Vector-borne diseases are commonly considered to be highly climate-sensitive, partially due to their explicit nature and monsoon climate, which fosters stagnant water [36,37]. There have also been reports of increased flooding along the eastern coast of India in recent years [33,38,39]. This flooding and water stagnation were mentioned by many participants as affecting their day-to-day life, including health. However, some participants mentioned that Puducherry is not severely affected by diseases like malaria, with a study finding a declining trend in malaria transmission in Puducherry between 2015 and 2021, with no observed seasonal trends [40]. Government-supported mass vector-borne disease awareness campaigns and vector control programmes over many years might have also contributed to cementing the stagnant water and vector-borne disease association in India, thereby lead to the strong interlining of climate change, floods and vector-borne diseases seen here [41,42].

We found a perceived need by participants to target interventions on multiple levels, taking into account the specific needs based on vulnerabilities, including socio-economic factors. The most economically disadvantaged are also the most vulnerable to exposure, which is likely to exacerbate in the coming decade primarily through health impacts, and as also affirmed by the participants, there is a need to target interventions at this group [43]. Climate-sensitive health risks disproportionately affect the most disadvantaged and vulnerable fractions of society, including women, the elderly and economically disadvantaged communities [13]. As highlighted by one of the participants, their needs have to be taken in account, especially when developing public infrastructure such as toilets, which are an important part of the political agenda of the Indian government [44].

Given the scale of recent agricultural damages due to extreme climate events in India, their widespread coverage and importance in the region, this was high on the list of perceived climate impacts. This included downstream effects of agricultural impacts on health through food systems, as was also seen in other multi-stakeholder studies from other settings, pointing to the interlinked nature of climate impacts [14,36,37]. As was also discussed by participants as an indirect impact of climate change, reduced agricultural yield due to climate change has been linked to a loss of work hours, which in turn can potentially lead to sedentary lifestyles, causing further risk of ill health and malnutrition [45,46].

Heat and season-related diseases including skin diseases, CVDs and respiratory illnesses, mentioned by few of the participants, are also commonly linked to climate change [16,37,47]. When it came to CVDs in particular, not all participants were convinced of the association. We found a gap in participants making the climate change-temperature-CVD connection, implying a need to improve awareness of the CVD or NCD impacts of climate change, which was also echoed by a participant. CVDs are mainly thought of as multifactorial and lifestyle-linked by all professionals, although several doctors appeared aware of how temperature affects the cardiovascular system. This was perhaps due to the tendency to disconnect temperature from climate change making it difficult for participants to connect it to the broader climate change manifestations, despite the growing number of studies conducted on the topic [45,48,49]. All participants also described being disconnected from the current climate-health research, and not having awareness of research and new developments by other national and international institutes. The need to improve inter-departmental communication and science dissemination in Puducherry is pronounced through the general dearth of awareness on the CVD impacts, compounded by the silent nature of the disease.

A need for improved awareness on climate change and health impacts emerged as a key finding, along with considerations on who delivers the message. Participants described a need to mobilize medical professionals in light of their emerging roles as trusted sources of information on health impacts of climate change and supporting public health policies, as has been observed in other global studies [17,50]. The high levels of public trust in physicians or perceived experts allows them to exert their influence and convince people easier than other professions, as was also alluded to by a participant in this study [50,51,52,53]. Celebrities as a source of knowledge, which we found in this study, have similarly been reported in another study from Puducherry [54]. Given that we found the political inclinations of information sources a contributing factor in their level of credibility, we can imply that the perceived ‘neutrality’ of the source is an important factor in how the information is received. However, based on our results, there is a need for doctors to receive more education and awareness on the topic, especially about CVDs and other non-explicit health impacts in order to confidently integrate climate change into their practice and disseminate information to the public [17,48,53]. Continued medical education courses are potential opportunities for the medical fraternity to recognise the health impacts of climate change and consciously implement it in their practice [17,55]. As a critical aspect of adaptation, education can serve as a major area through which to implement behavioural change, especially if started at a young age [56]. With respect to that, there is also a need to assess the level of knowledge, misconceptions and confidence in explaining climate change concepts among teachers, as also demonstrated in a Canadian study [57].

We found that there remains a need to provide alternate solutions and target design interventions specific to different groups in Puducherry, starting from the ground level, which has been shown to increase public support for actions [58,59]. Targeting the most vulnerable groups, such as farmers or those exposed to the external environment for an extended time was considered to be important among study participants. According to the participants, the DSTE has already made first steps in this direction with their tailored programmes to specific groups like farmers and children. However, there remains scope to include health impacts of climate change in these programmes which serve as an important mode of accessible science communication. The media has been proven to be another powerful source of information, one which most people trust and consider reliable [16]. In fact, participants in this study mentioned it as being the most common and influencing source, showing the great scope for utilizing it to further the agenda and spread the message among common people.

Participants also mentioned the need for occupational health protection measures as a response to climate change, especially among those engaged in outdoor and manual labour. Exposure to heat is a major driver of occupational vulnerability due to heat stress and it affects worker health through increased vulnerability to diseases such as CVDs, and productivity [60]. In India, common reasons for reduced work capacity included exhaustion, hospitalization and lost wages, which also greatly reduce the economic and development pace and capacity of the country [61]. Worker productivity decreases when the Wet Bulb Global Temperature (WBGT) is above 26 to 30 °C [62]. Puducherry, with an average temperature of 30 °C, has an apparent temperature range between 23 and 41 °C, meaning it is especially important to protect the health of outdoor workers such as farmers, fishermen, construction and manual labourers [63,64]. Policy measures to reduce workplace heat exposure in the context of climate change and improve occupational health are imperative to protect the working population, as has also been recommended by WHO [60,65]. Similarly, it is essential to plan actions with the aim of protecting children’s health in the context of exposure to heat in schools [66,67].

This study has some limitations, which we present here for consideration. First, our study was small in scale and was of an exploratory nature, restricted to Puducherry. Second, despite the saturation reached in the information, further studies with a larger sample size might prove to be advantageous. Third, there was a gender imbalance in the study, with only very few female participants, especially regarding medical professionals. Fourth, our study was limited only to health professionals and environmentalists in the policy sphere, and hence, it did not include the views of the broader public. In particular, we recommend future studies on the topic to include the perspectives of the most vulnerable and marginalized population to gain a deeper understanding of the climate risks they face as well as areas for improving interventions.

## 5. Conclusions

There are limited data on how climate change and its health risks are perceived by key stakeholders, especially in the LMICs. Our study sheds new light on this topic among medical professionals and environmentalists in the Puducherry region as well as some perceived gaps and recommendations. While there was awareness on the impacts of climate change, including on health, we found a disconnect when it came to diseases not conventionally associated with climate, such as CVDs. There is scope for education and training, especially among healthcare providers, on climate-sensitive diseases. We also report a high perceived need for more education and awareness on climate change and health, not only among the scientific community but also among the general population, along with recommendations for adaptation measures. It also highlights the need for studies in other regions of India and elsewhere. These findings can be used to strengthen the region’s climate action plan through targeting key areas identified in this study such as education and awareness building on health impacts of climate change, not only among the local population but also among key stakeholders.

## Figures and Tables

**Figure 1 ijerph-20-04703-f001:**
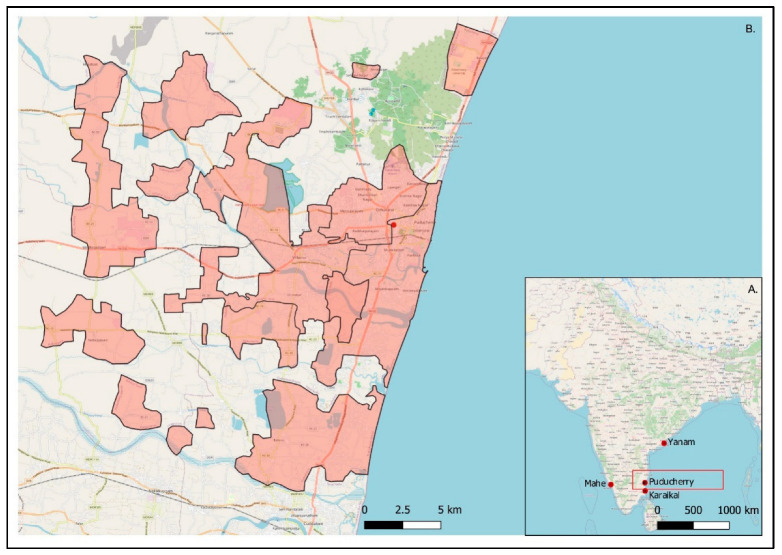
Map of Puducherry. (**A**) Location of the four districts that make up the Union Territory of Puducherry, namely Puducherry, Karaikal, Mahe and Yanam, spread out on either side of the coast of India. (**B**) Puducherry district, which is nestled within the state of Tamil Nadu with Andra Pradesh to the north (inlaid map). The shaded area highlights the non-continuous geographical area of Puducherry district.

**Figure 2 ijerph-20-04703-f002:**
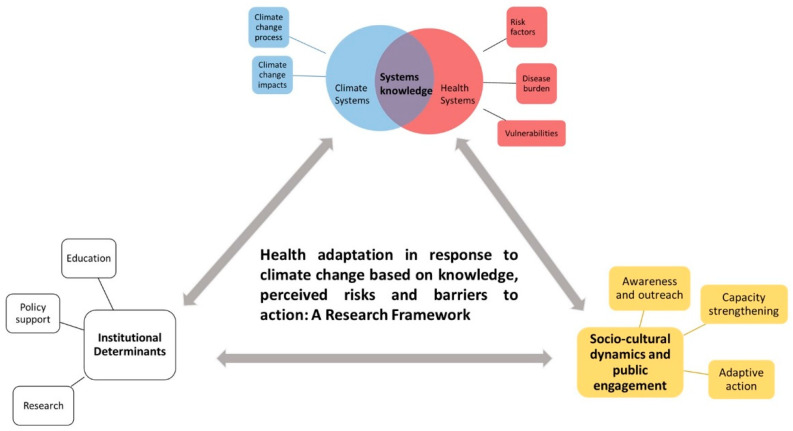
A framework for health adaptation action in the context of climate change based on knowledge and perceived health risks, policy and institutional support and public engagement. The coloured parts highlights the thematic areas we focus on in this paper. The three main components have been shown in bold text.

**Table 1 ijerph-20-04703-t001:** Descriptive characteristics and participant profiles of the key informants interviewed in this study.

Sector/Background	*N* (%)	Females (%)	Males (%)	Age Range (Years)	Range of Experience (Years)
Medicine (in-practice)	8 (50%)	0 (0%)	8 (50%)	32–51	3–20
Medicine (research/academic)	3 (18.8%)	1 (6.3%)	2 (12.5%)	40–44	11–20
Environment/governmental	5 (31.3%)	2 (12.5%)	3 (18.8%)	28–53	4–30
Total	16	3 (18.8%)	13 (81.3%)	28–53	3–30

## Data Availability

All relevant data from this study have been included in the Appendix A. As this is a qualitative study with a small number of key informants, making the full dataset and interview transcripts available to a wider audience could potentially breach the confidentiality commitment made to the participants during the process of obtaining informed consent as well as to the ethics committees that approved this study. Therefore, the data will not be made available.

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
