# Peer review of "“Climate Change and Health?”: Knowledge and Perceptions among Key Stakeholders in Puducherry, India"

_ijerph, 2023, doi:10.3390/ijerph20064703_

Round 1

Reviewer 1 Report

Comments to author

The authors approached a significant investigation field “  Climate change and health?”: Knowledge and perceptions among key stakeholders in Puducherry, India”. The article can be interesting for the readers of the International Journal of Agronomy and is suitable for publication in the journal after minor changes.

General Comments.

The authors tried to evaluate the  effects of Climate change and health however, introduction section need some improvement. Authors should dicuss the effect of climate change on crop productivity globally ,specifically Asian countries

Introduction.

1.     The introduction section need improvement. Add some more citations from the recent literature

Material and Methods.

2.     Line # 135. A total of sixteen interviews (n=16) were conducted ( Does the authors think that only sixteen samples are anough to justify their hypothesis ? Please explain.

3.     Figure 2: Please upload a clear image with better resolution.

References.

4.     Please add some more citations from most recent studies to  strengthen the results. 

Reviewer 2 Report

A review on the manuscript in International Journal of Environmental Research and Public Health entitled „“Climate change and health?”: Knowledge and perceptions among key stakeholders in Puducherry, India“

Using qualitative research, the article mainly discusses interviews with sixteen doctors, scientists, environmentalists and government officials who worked on the Climate Change Action Plan for Puducherry, India. The results have been analyzed using a framework method with data-based thematic analysis.

The results of the study, based on a limited dataset, are adequate and correspond to the results published in the article.

Unfortunately, the presentation of data does not follow the basic principles of academic writing.

The article needs to be significantly improved in order to be published.

Broad comments

Academic writing should be objective. If it is subjective or emotional, it will lose persuasiveness and may be regarded as relying on emotion rather than building a reasonable argument based on evidence. The language or informal writing should therefore be impersonal, and should not include personal pronouns. For most subject areas the writing is expected to be objective. For this the first person (I, we, me, my, etc.) should be avoided. In this article on line 17 is written „We conducted“, on line 20 is written „We elucidated“, on line 34 is written „ we interviewe“, on line 35 is written „we cannot“, on line 81 is written „we aimed“, on line 83 is written „we discussed“, on line 84 is written „we chose“, on line 109 is written „We conducted“, on line 115 is written „We specifically chose“, on line 116 is written „we were“, on line 118 is written „ We continued“, on line 119 is written „we had“, on line 124 is written „we focused“, on line 150 is written „we focus“, on line 173 is written „We used“, on line 174 is written „ We first developed“, on line 370 is written „we wanted“, on line 371 is written „We found“, on line 560 is written „we attempted“, on line 588 is written „We found“, on line 609 is written „We found“, on line 629 is written „we found“, on line 630 is written „we found“, on line 631 is written „we can“, on line 639 is written „We found“, on line 676 is written „we present“, on line 680 is written „we recommend“, on line 691 is written „we found“, on line 693 is written „We also report“. Eliminating personal pronouns from writing is highly recommend.

The article needs a thorough technical correction.

Specific comments

In the article, the subheadings are not formatted in the same style, for example, there is a dot after the number of the subheader 3.1. and 3.2., but there is no dot in the subheadings2.1, 2.2 etc, etc.

The numbering of the sub-headings of the article needs to be arranged, for example there are two sub-chapters 2.4 - line 170 and line 192.

Each figure must have a capture, which is under Figure. Unfortunately, this is not the case with Figure 1.

Each sentence must end with a period. Unfortunately, this is not the case on lines 224, 231, 229, 248, 257, 269, 281, 294, 316, 332, 344, 367, 392, 396, 403, 423, 440, 455, 478, 493, 528 and 550.

All paragraphs should be formatted in the same style in the article. But on lines 370 to 423, the paragraphs are not formatted with indentation.

Reviewer 3 Report

The manuscript entitled " “Climate change and health?”: Knowledge and perceptions among key stakeholders in Puducherry, India " is focusing mainly on CVDs as well as other diseases and their association with climate change, by going through a qualitative study using key informant interviews where the research gap has been properly identified. Generally, the manuscript is well structured, the sections are well organized and clear. That said, the manuscript lacks certain information.

The novelty of the study is missing. Authors can include it in the introduction.

The methodology must include relevant details, and further information about the interviews :

1. How was the purpose of the study explained to the participants before starting the interview ?

2. Were the recordings given to the participants ? Did any of the participants give a feedback about the study ?

3. Transcription was done by who ?

4. The paragraph from line 418 to line 423 : Does the quote belong to a practicing physician or a cardiologist ? Please specify this detail in line 416.

5. Line 670 : Authors must change the citation "[59]", as it is not quite applicable.

6. The paragraph from line 652 to line 659 : I highly invite the authors to consider this paragraph. Various information regarding the participant's children was never mentionned. What's their age range ? In which educational stage these children are ? How many participants has declared to have children ? Climate change education may be limited in Puducherry schools. However, based on the information provided by the participants, authors are not able to draw such conclusions in this regard.

7. The quotes in Supplementary files should include the profile of the participant, as adopted in the main manuscript.

8. I suggest adding the pariticipants' contribution to better assess the understanding of the topic among stakeholders.

Reviewer 4 Report

This study discussed the relationship between climate change and health from the perspective of stakeholders, using Puducherry, India, as an example. I review this manuscript due to a strong interest in the topic of “climate change and health”. I have quite a few suggestions.

Although the description in the manuscript emphasizes data-driven analysis, it seems that the manuscript did not seem to address the data analysis component. The results of the manuscript seemed to be based on mere excerpts from the interviews, which could very easily be subjective.

The current version of the manuscript did not seem to bring out the real point that needs to be made. I believe that the entry point in the introduction and corresponding results should be directed towards specific findings, including the phenomenon and the explanations behind it, rather than just stating the direct results of the interviews.

The number of people interviewed was too small, only 16. Such a small sample size may make it difficult to obtain objective results. In addition, I believe that the proportion of female respondents is significantly lower compared to men, which may affect the diversity of the interview results and thus the findings in the study.

Why were the interviewees divided into medical professionals and environmentalists? The manuscript did not show differences in the results or highlight the particular categories of interviewees. In fact, there are three categories in Table 1, which need to be analysed in the results and discussion. At the same time, the sample size of medical professionals is much larger than that of environmentalists, making comparisons of results between categories inappropriate in terms of statistical significance.

Why was the region of Puducherry chosen for the interviews? What are the characteristics of this region in terms of climate change or in terms of health? How is the relationship between climate change and health in this region typical of India or globally, and what are the particular implications of the dispersion of the Union Territory of Puducherry for the relationship between climate change and health?

I noted significant variation in the timing, location, and length of interviews conducted by interviewees. These differences can directly influence the results of the interviews, especially if the sample is too small.

Although the description in the manuscript makes CVDs the focus of the analysis, the results section did not seem to bin around this topic

Result sections 1 and 2 appear to be somewhat disconnected. It is recommended that the logical link between Result 1 and 2 be enhanced. From the current results it is difficult to understand what the authors are currently trying to say.

Round 2

Reviewer 2 Report

A review on the manuscript in International Journal of Environmental Research and Public Health entitled „“Climate change and health?”: Knowledge and perceptions among key stakeholders in Puducherry, India“

Using qualitative research, the article mainly discusses interviews with sixteen doctors, scientists, environmentalists and government officials who worked on the Climate Change Action Plan for Puducherry, India. The results have been analyzed using a framework method with data-based thematic analysis.

The results of the study, based on a limited dataset, are adequate and correspond to the results published in the article.

Unfortunately, the presentation of data does not follow the basic principles of academic writing.

The article needs to be significantly improved in order to be published.

Broad comments

Academic writing should be objective. If it is subjective or emotional, it will lose persuasiveness and may be regarded as relying on emotion rather than building a reasonable argument based on evidence. The language or informal writing should therefore be impersonal, and should not include personal pronouns. For most subject areas the writing is expected to be objective. For this the first person (I, we, me, my, etc.) should be avoided. In this article on line 17 is written „We conducted“, on line 19 is written „We elucidated“, etc., etc. Eliminating personal pronouns from writing is highly recommend. Unfortunately, this recommendation has not been heeded in the slightest.

Author Response

Dear Reviewer,

Thank you for your comment. Regarding the use of personal pronouns, we understand your point and, as we mentioned, while we agree that emotional writing should be avoided, we do not think it conducive to use the passive voice, especially since this is purely qualitative research. In qualitative methodology, the researcher is very much part of the data collection and analysis, as we have detailed in the methods section. We feel using the passive voice would not only introduce a positivist approach to the results, which we do not want, but also hamper the natural flow and clarity of the work. In addition, majority of qualitative research papers are written using personal pronouns, including papers in the IJERPH. Thus we feel our preference to use the active voice is within the bounds of both the editorial policies at the journal and the common writing style for qualitative research.

We found your overall comments extremely helpful and thank you again for your work in improving our paper.

Reviewer 4 Report

Thanks for your work.

Author Response

Dear Reviewer,

We thank you again for taking the time to improve our work.